# Efforts and Challenges in Engineering the Genetic Code

**DOI:** 10.3390/life7010012

**Published:** 2017-03-14

**Authors:** Xiao Lin, Allen Chi Shing Yu, Ting Fung Chan

**Affiliations:** School of Life Sciences, The Chinese University of Hong Kong, Sha Tin, NT, Hong Kong, China; xlin@link.cuhk.edu.hk (X.L.); allenyu@cuhk.edu.hk (A.C.S.Y.)

**Keywords:** frozen accident, genetic code, genetic engineering, evolution, synthetic biology

## Abstract

This year marks the 48th anniversary of Francis Crick’s seminal work on the origin of the genetic code, in which he first proposed the “frozen accident” hypothesis to describe evolutionary selection against changes to the genetic code that cause devastating global proteome modification. However, numerous efforts have demonstrated the viability of both natural and artificial genetic code variations. Recent advances in genetic engineering allow the creation of synthetic organisms that incorporate noncanonical, or even unnatural, amino acids into the proteome. Currently, successful genetic code engineering is mainly achieved by creating orthogonal aminoacyl-tRNA/synthetase pairs to repurpose stop and rare codons or to induce quadruplet codons. In this review, we summarize the current progress in genetic code engineering and discuss the challenges, current understanding, and future perspectives regarding genetic code modification.

## 1. Introduction

In 1968, Francis Crick first proposed the frozen accident theory of the genetic code [1]. The 20 canonical amino acids were once believed to be immutable elements of the code. The genetic code appears to be universal, from simple unicellular organisms to complex vertebrates. Yet in contrast to studies of the natural selection of lifeforms wherein the gradual evolution of species can be observed in a myriad of taxa, relatively few examples of natural genetic code variations (e.g., selenocysteine [2], pyrrolysine [3,4], and stop codon read through [5,6]) have been observed. Different explanations have been proposed to address these variations, such as the codon capture hypothesis [7], the ambiguous intermediate hypothesis [8], and the genome streamlining hypothesis [9]. These hypotheses have been reviewed elsewhere [10,11]. Although some existing noncanonical amino acids (NCAAs) are known to be compatible with enzymatic aminoacylation [12,13,14,15,16,17,18,19,20], the 20 canonical amino acids in the standard genetic code have been stringently selected over the course of biological evolution. Organisms that require peptides with modified side chains will often resort to pre-translational or post-translational modifications to incorporate NCAAs [21,22,23,24,25,26]. Some organisms require alternative genetic codes to survive in harsh living conditions [27]. 

Genetic code engineering refers to the modification, or the directed evolution of cellular machineries, in order to incorporate NCAAs into the proteome of an organism. In general, NCAAs can be artificially incorporated in a site-specific or proteome-wide manner. In the former, scientists have attempted to artificially engineer organisms for compatibility with various NCAAs by employing orthogonal tRNA/aminoacyl-tRNA synthetase pairs [28,29,30,31,32]. In the latter, an organism is forced to take up specific NCAAs, followed by isolating mutants in media containing NCAAs [33,34,35]. Currently, researchers are recording cellular responses and genetic changes in engineered organisms to understand the mechanisms behind the use of alternative genetic codes. Such efforts enable an understanding of the evolutionary course of the genetic code and provide a foundation for the derivation of additional alternative codes, a particularly important feature in the era of synthetic biology given the increased focus on engineering synthetic organisms with modified genetic codes [36,37]. Engineered genetic code holds tremendous potential in the field of protein engineering and xenobiology, which was extensively reviewed by Budisa et al. in 2017 [38].

In this review, we first give a brief introduction of current studies on both site-specific and proteome-wide incorporation of NCAAs. Next, we will focus on the challenges of engineering organisms to use modified genetic codes and their implications, such as inhibitory effects caused by NCAAs. Finally, we will discuss current trends in this research area.

## 2. Genetic Code Engineering

### 2.1. Incorporation of NCAAs into Specific Sites

Currently, three major approaches are used to engineer the genetic code in a site-specific manner: (1) amber codon suppression; (2) rare sense codon reassignment; and (3) quadruplet codon. Figure 1 provides a schematic illustration of each method. Because organisms such as *E. coli* BL21 rarely use the amber stop codon (UAG) (only 275 of 4160 stop codons in BL21 are amber codons), which minimizes disturbances to existing protein termination signals, this codon has been preferably selected for NCAA encoding [30,39,40]. To enhance the efficiency of amber codon recognition by the orthogonal tRNA_CUA_, Release Factor 1 [41,42] is usually mutated or knocked out [43], thus enabling orthogonal tRNA_CUA_ to recognize and increase its competitive binding to the amber codon [30] (Figure 1a). The role of different artificial tRNA/tRNA synthetase pairs, as well as their structural relationship with different NCAAs, were extensively reviewed by Anaëlle et al. [44].

Rare sense codon assignment [45,46,47], which is based on a similar principle, repurposes rare sense codons, particularly rare codons including AGG [45,46] and AUA [47], using newly designed tRNA/aminoacyl-tRNA synthetase pairs. In this method, the introduction of a NCAA during protein synthesis requires either competition between a genetically modified tRNA and the corresponding wild-type tRNA [45] or the inhibition of wild-type tRNA via the deletion of its tRNA synthetase [47] (Figure 1b).

To circumvent the limitations of reprogramming existing codons, some researchers have explored NCAA encoding via expansion of the genetic code using quadruplet codons [48,49,50,51,52,53]. In brief, a single-base (e.g., “U”) is inserted after a canonical triplet codon (e.g., a “CUC” triplet codon) to form a frameshift mutation at the specific position (Figure 1c). The additional base also creates a new quadruplet codon (e.g., “CUCU”) at this position, which can be recognized by an engineered quadruplet tRNA (e.g., tRNA_AGAG_). Early versions of the quadruplet in vivo coding system were initially tested in *E. coli* [48,51,52], followed by *Xenopus* oocytes [49] and mammalian cells [50,53]. It is also worth mentioning that noncanonical RNA translations, such as the use of tetra- and penta-codon, were observed in mitochondria; however, the 4th and 5th nucleotides were found to be silent during translation [54]. More mechanistic studies would be required to establish their roles in genetic code engineering.

Although the site-specific incorporation approach is arguably the most widely used to produce artificial proteins with NCAAs, some challenges can limit the stability of the engineered code. The efficiency of an engineered tRNA/aminoacyl-tRNA synthetase pair must be high enough to minimize the generation of truncated proteins [55]. Methods such as orthogonal ribosome use can lead to a threefold improvement in the efficiency of unnatural amino acid incorporation [55]. Endogenous tRNA/aminoacyl-tRNA synthetase pairs can also be engineered to incorporate unnatural amino acids. For example, by changing the phenylalanyl-tRNA synthetase amino acid recognition site, phenylalanine analogs such as p-Cl-phenylalanine or p-Br-phenylalanine can be successfully charged to tRNA^Phe^ [14,56]. Advances in genome editing techniques, such as multiplex automated genomic engineering [31] and CRISPR/Cas [51], may further increase the efficiency and accuracy of NCAA incorporation in specific sites of the proteome. 

### 2.2. Proteome-Wide Incorporation of NCAAs

Proteome-wide incorporation offers an alternative approach toward unnatural amino acid incorporation. In the most common approach, amino acid uptake is artificially controlled by feeding auxotrophs with NCAAs [33,34,35] (Figure 2a). Attempts to control NCAA synthesis have involved supplying organisms with NCAA precursors [57,58,59,60,61,62] (Figure 2b), which is also known as metabolic engineering [63,64]. In one example, the precursor l-β-thieno [3,2-b]pyrrolyl ([3,2]Trp) was fed to a tryptophan (Trp)-auxotrophic *E. coli* capable of synthesizing [3,2]Tpa (a Trp analog) to generate mutants that could propagate on l-β-(thieno [3,2-b]pyrrolyl)alanine ([3,2]Tpa) [57] (Figure 2b). Although directly feeding auxotrophs with NCAAs is a simpler approach, metabolic engineering could reduce the unwanted effects of impure commercial NCAAs [57].

Regardless of approach, the incorporation of NCAAs in the proteome may negatively affect the growth of an organism. The inherent toxicities of many unnatural amino acids could suppress propagation of the wild-type strain and select mutants that respond favorably to the NCAA, ultimately causing rejection of the expanded genetic code [34,35]. In the following section, we will focus on the challenges in genetic code modification.

## 3. Challenges of Genetic Code Engineering

### 3.1. Inhibitory Effects of Engineered Genetic Codes

The growth inhibitory effects caused by NCAAs, which have been demonstrated in different species including bacteria [65,66,67], yeasts [68], insects [69,70], and mammals [71], comprise one major challenge encountered during genetic code modification. The inhibitory effects of NCAAs are mainly attributable to two aspects. First, minor structural and chemical differences between NCAAs and their canonical counterparts can drastically affect enzymatic activities [72,73,74,75]. Second, these structural and chemical differences may also negatively affect protein synthesis, as some NCAAs cannot be efficiently charged to tRNAs by aminoacyl-tRNA synthetases [15,76]. A better understanding of the key genes and cellular responses associated with these modified genetic codes is of paramount importance to alleviating these inhibitory effects.

### 3.2. Discovering the Key Genes Controlling the Genetic Code

The growth inhibitory potentials of NCAAs create negative selective pressure, while the organism adapts to the modified genetic code. One effective strategy for overcoming this evolutionary barrier comprises an increase in the mutation rate via mutagenesis with the expectation of generating beneficial mutations that would favor the NCAA. Wong and colleagues isolated mutants from a Trp auxotroph (*Bacillus subtilis* str. QB928) via sequential mutagenesis in an early attempt to modify the genetic code. The resultant HR23 strain could propagate indefinitely on 4-fluoro-tryptophan (4FTrp) but became inviable on canonical Trp [34,35]. As Trp is encoded by a single codon (UGG), the research by Wong and colleagues provided the first evidence of codon membership malleability under external selection pressure. Subsequently, Yu et al. traced mutations in intermediate mutants, as well as the HR23 strain [77]. A nonsense mutation in the Trp operon RNA-binding attenuation protein (TRAP), which controls transcriptional attenuation of the Trp operon [78,79] and translational repression of Trp transporters [80,81,82,83], was shared by all mutants. This lack of TRAP would increase 4-FTrp uptake to compensate for the relatively low charge rate of 4-FTrp to tRNA^Trp^ [15].

In a separate attempt, Bacher et al. isolated *E. coli* mutants that could propagate in medium wherein 4-FTrp comprised ~99% of available Trp. However, the mutant strains could not grow indefinitely under these conditions and required minimal canonical Trp [33]. *E. coli* mutants were found to harbor several mutations affecting genes such as *aroP*, which encodes an aromatic amino acid transporter [84], and *tyrR*, which encodes the associated regulator [85]. Mutated *aroP* and *tyrR* might cooperatively increase 4-FTrp uptake, similar to the effect of TRAP knockout in *B. subtilis.* Taken together, these findings suggest that an efficient NCAA uptake system is essential to accommodation of the modified genetic codes. 

RNA polymerase might also play a key role in controlling the genetic code. The above-mentioned *B. subtilis* mutant HR23 was found to harbor a nonsynonymous mutation in the RNA polymerase subunit gene (*rpoB*) that was absent from all other intermediate strains that could still propagate on Trp, suggesting a potential role for this mutation in switching membership of the UGG codon from Trp to 4-FTrp [77]. In an independent study of amber codon-directed 3-iodotyrosine (3-iodoTyr) incorporation in *E. coli*, a *rpoB* mutation was found to confer rifampicin resistance via amber suppression at Gln513 [86], and the same research group also engineered a bacteriophage, T7, that could incorporate 3-iodoTyr at amber codons [29]. In that study, Hammerling et al. observed high mutation frequencies in genes encoding RNA polymerase and the lysis timing regulator type II holin. The authors suggested that these two genes played important roles in the evolution of the expanded genetic code [29]. These studies have shed light on the previously unexplored roles of key genes in genetic code identity.

### 3.3. Lack of Transcriptomic and Proteomic Studies Related to Engineered Genetic Codes

In addition to mutations, gene and protein expression profiles might also reveal key factors needed to fine-tune the use of modified genetic codes. Technologies such as RNA-seq and mass spectrometry can be used to investigate the cellular responses of organisms in high resolution. RNA-seq was used to compare the cellular responses between mutant (grown on 4-FTrp) and wild-type strains of the above-mentioned *B. subtilis* HR23 mutant (unpublished data). Here, a gene ontology analysis of the gene expression profiles of these strains demonstrated enrichment of genes related to reactive oxygen species responses and branched-chain amino acid biosynthetic processes among upregulated genes, and enrichment of genes related to siderophore biosynthetic processes among downregulated genes (unpublished data). Unsurprisingly, stress response genes were modulated in response to the new genetic code, and the downregulation of siderophore biosynthetic process related genes was consistent with a previous observation of the reduced growth rate of HR23 cells grown on 4-FTrp [77] because iron homeostasis is closely related to bacterial growth [87]. This unique set of data was the first to demonstrate the adaptation of an organism to a new genetic code at the transcriptomic level.

*Methanosarcina acetivorans* is a methanogenic archaea strain that uses the alternative genetic codon UAG to encode pyrrolysine (Pyl) [88]. O’Donoghue et al. attempted to reduce the genetic code of this strain by deleting tRNA^Pyl^, thus blocking the incorporation of Pyl in the proteome. A comparison of the proteomes of mutant and wild-type *M. acetivorans* strains revealed that most upregulated peptides were related to methanogenesis, protein synthesis, and the stress response [89], suggesting that, in this organism, various stress response genes must be fine-tuned before a reduced genetic code can be used.

Very few transcriptomic and proteomic studies of organisms with modified genetic codes have been conducted, and we have only glimpsed the potential factors involved in adaptation to modified genetic codes. Additional genes that contribute to this adaptation might remain to be discovered. In the future, studies of gene and protein expression in organisms with modified genetic codes will be necessary.

### 3.4. Environmental Factors Affecting Adaptation to Engineered Genetic Codes

Environmental factors, such as the growth medium and selection method, are important when optimizing the use of a modified genetic code. The amino acid source is the first and most obvious factor, as an organism can either take up NCAAs directly from the environment or synthesize them using environmentally available molecules. If the source of NCAAs is from the environment, mutations in amino acid transporters are often needed to facilitate NCAA uptake [33,77].

Positive selection pressure is also needed to maintain stability of the modified genetic code. In a previous study, incorporation of the methionine analog azidohomoalanine (Aha) into the coat protein of a human adenovirus and the subsequent addition of a folate group to Aha facilitated adenoviral infection in mouse hosts [90]. In other words, adenovirus strains that can use modified Aha have a selective survival advantage over other strains. In a more recent study of different *E. coli* strains, the site-specific incorporation of two tyrosine analogs in β-lactamase was selected, and enzymatic function was found to depend on the presence of these analogs [91]. As described above regarding adenovirus, *E. coli* mutants that could utilize NCAAs enjoyed a selective advantage under growth mediums containing certain classes of antibiotics [91]. In one interesting example, even the carbon source may affect the selection of genetic codes by the Pyl-utilizing bacteria *Acetohalobium arabaticum* [92]. *A. arabaticum* used the standard genetic code when grown on pyruvate, but gained the ability to use an expanded genetic code that included Pyl in the presence of the alternative carbon source trimethylamine [92].

## 4. Future Directions

Current efforts in genetic code engineering have reshaped our ideas regarding genetic code evolution and have paved the way for expanding the genetic alphabet. Based on these studies, we have outlined the key steps by which an organism accommodates a modified genetic code (Figure 3). During adaptation, mutations in amino acid transporters and/or their key regulators allow more efficient NCAA uptake, possibly by increasing the number of amino acid transporters [33,77]. Mutations in the key genes might also favor the use of a modified genetic code [29,33,77,86]. Additionally, environmental positive selection forces contribute to stability of the modified genetic code [77,91,92]. Currently, the genomic changes in organisms with modified genetic codes have been well explored [29,33,77,86] relative to transcriptomic (unpublished data) and proteomic [89] changes. Future trends in elucidation of the biological mechanisms underlying genetic code modifications include the integration of genomic, transcriptomic, and proteomic data and the refining of functional study targets.

From the viewpoint of synthetic biology and xenobiology, genetic code engineering increases the repertoire of building blocks available for protein engineering, thus enabling the development of novel proteins that would be impossible with canonical amino acids [38,93]. Xenobiology is an emerging field that involves synthesizing xenonucleic acids other than the canonical nucleic acids with adenine (A), thymine (T), cytosine (C), and guanine (G) as bases, with alternative pairing rules for protein engineering [94]. It has been demonstrated experimentally that two such xenonucleic acids can be integrated into the current DNA backbone [95,96,97], and more have been tested for their potentials as novel building blocks of DNA [98]. With the addition of xenonucleic acids, the number of encoded amino acids is likely to be increased to far beyond 20 [94].

High-throughput genome editing technologies, such as MAGE [36] and the emerging CRISPR/Cas technology [99], allow an organism’s genetic code to be directly rewritten [37,100] and facilitate the creation of synthetic life [101]. Although the first synthetic minimal bacterial genome still uses the standard genetic code [101], it is now possible to synthesize genomes based on alternative genetic codes. A full exploration of the possibilities enabled by genetic code engineering requires an understanding of the key molecular biological and biochemical mechanisms underlying the modifications. Gradual efforts to address this main question may improve our understanding of the process of genetic code evolution and lay a better foundation for future synthetic biology research.

## Figures and Tables

**Figure 1 life-07-00012-f001:**
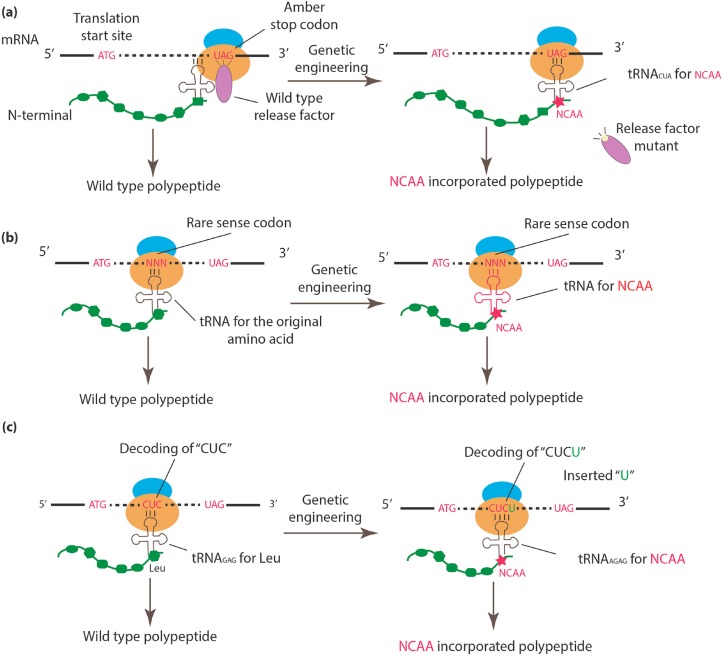
An overview of approaches to incorporate NCAAs into specific sites. (**a**) The wild-type release factor is mutated or knocked out, allowing the newly introduced tRNA_CUA_ to read through the stop codon, followed by NCAA incorporation with assistance from the compatible aminoacyl-tRNA synthetase. (**b**) The tRNA and corresponding tRNA synthetase for a rare sense codon are genetically engineered to confer the ability to encode NCAA. (**c**) A single-base is inserted after the canonical codon (e.g. “CUC” for Leu). The newly introduced quadruplet tRNA (e.g., tRNA_AGAG_) can encode NCAA by targeting the quadruplet codon “CUCU.”

**Figure 2 life-07-00012-f002:**
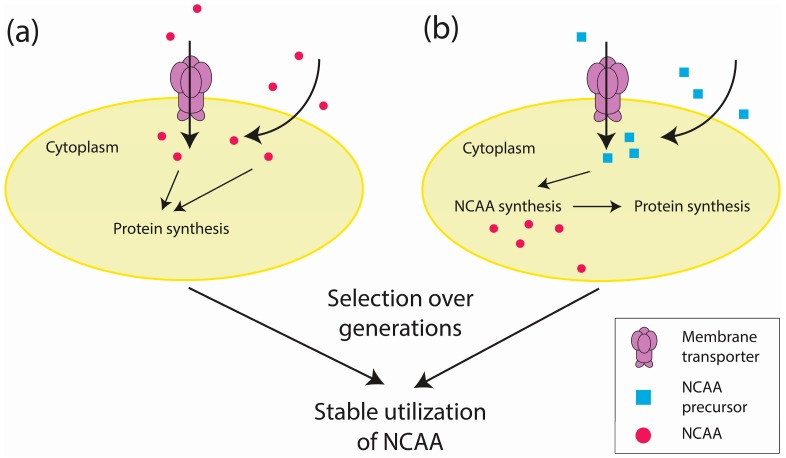
An overview of proteome-wide approaches to incorporate NCAAs. (**a**) The NCAA enters a cell via membrane transporters or diffusion across the membrane. (**b**) The NCAA precursor similarly enters a cell in which it will be used to synthesize NCAAs. Following several generations of propagation with either the NCAA or its precursor, cells that can stably utilize the NCAA are selected.

**Figure 3 life-07-00012-f003:**
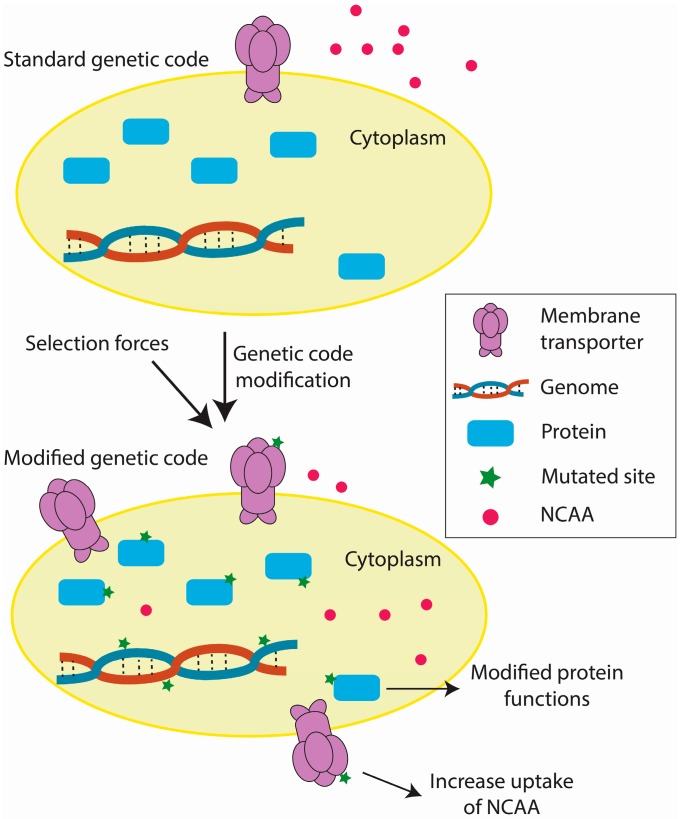
Key steps in the accommodation of a modified genetic code.

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
