# Peer review of "Efforts and Challenges in Engineering the Genetic Code"

_life, 2017, doi:10.3390/life7010012_

Round 1
Reviewer 1 Report
This manuscript is well-written about recent progress on genetic code expansion in living cells. I would like to agree to accept the manuscript, if the following points are appropriately revised.
Line 60:
The authors describe "release factor 1 must be mutated or knocked out" for amber suppression. However, amber suppression can usually be achieved in wild-type cells, and therefore, mutation or knockout of release factor 1 is not essential for amber suppression.
Line 182:
"M. acetivorans" should be "Methanosarcina acetivorans".
References 31 and 61:
There are no journal titles.
Reference 72:
This reference seems to be a PhD thesis of one of the authors. I recommend that peer-reviewed paper should be cited. If there is no peer-reviewed paper, the authors should clarify this is unpublished result.
Author Response
We thank the reviewer for reading the manuscript carefully. The problems spotted in line 182, which is in line 186 in the revised version, and in the reference have been fixed. For the issue in line 60, which is in line 64, we have rephrased the sentence according to the suggestion.
Reviewer 2 Report
The review manuscript of Lin et al covers interesting recent developments in the research field known as “genetic code expansion” (GCE) and relate them with the “frozen accident” hypotheses of Francis Crick postulated already in 1968.
Immediately after genetic code elucidation, its universality in all organisms becomes obvious which probably motivated Crick to suggest that current code is the result of a “frozen accident“. It means, that genetic code was unable to evolve further even if the current state were suboptimal. His main argument was that “no new amino acid could be introduced without disturbing too many proteins” - indicating that current code is no more evolvable.
Crick was thinking about proteomes, not single proteins. Whereas “thawing” the genetic code for a single recombinantly expressed proteins is not big deal anymore, proteome wide codon reassignments are really challenges – only with these achievements “thawing the frozen landscape” would be possible.
Unfortunately, the authors a mixing these issues through the whole manuscript. For example, already in the abstract it is stated that:
“Currently, successful genetic code modification is mainly achieved by creating orthogonal aminoacyl-tRNA/synthetase pairs to repurpose stop and rare codons or to induce quadruplet codons.”
This is all done in the frame of single recombinantly (normally plasmid-encoded) expressed proteins. It has nothing to do with “thawing the frozen landscape”. It can be only first step in this direction (in the best case).
So far, only experiments reported by Wong (1983) and Ellington (2001) represents first attempts to change amino acid repertoire of the genetic code by replacing Trp with fluorotryptophans. However, these experiments never yielded (analytically proved) full replacement of Trp (as it is always present in traces in commercially prepare amino acids). Only recently, Budisa recently succeeded in full replacement of Trp with sulfur containing theinopyrole-alanine (by presenting clear cut analytics; proteome and other analyses will most probably be published soon).
Obviously, the authors should define what is “thawing” the genetic code. It is not codon reassignment on single proteins in the frame of routine recombinant DNA technology! It is proteome-wide amino acids (NCAA) exchange, additions or reductions (see http://www.eurekaselect.com/122762/article ) as a result of genome-wide codon reassignments! This is also only acceptable criterion for “thawing the frozen landscape” (as suggested above) since it is really related to the original Cricks idea of “frozen accident” from 1968.
In 2006, RajBhandary and Soll published a paper “The genetic code – Thawing the ‘frozen accident’” (not cited by the author of this manuscript!) where they were considering following facts. First, the genetic code is the product of its interpretation by the translational machinery and it is only static as long as the component of this machinery do not change/evolve. Second, a diverse assortment of alternation of the genetic code have now being documented, mainly involving stop codons (in organelle but also in some nuclear genes). Third, these variations are explained by “codon capture” and “ambiguous intermediate” theories (not even mentioned by the author of this manuscript!) and are related (among other) to tRNA post-transcriptional modifications. Fourth, pyrrolysine and selenocystene are natural example of GCE achieved by stop codon reassignments.
Unfortunately, the authors field to provide a short and comprehensible overview to get an idea about theory of codon reassignment – a crucial event in any GCE experiment in synthetic biology. Therefore, they are strongly suggested to look (and cite) at least some recent papers where these issues (and their relevance for GCE) are discussed more detail (see for example: https://www.ncbi.nlm.nih.gov/pmc/articles/PMC4993186/ ).
Finally, genetic code redesign or modification also include work on noncanonical DNA (and even RNA) structures - and authors are strongly suggested to mention this at the end of the introduction section of their revised manuscript. To get an idea, they can look at the Fig. 1 of the recently published essay of Acevedo-Rocha (https://www.ncbi.nlm.nih.gov/pmc/articles/PMC4993186).
After these clarifications are performed – current terminology of the paper need to be systematically purged.
The term “Bottom up approach” is wrong in this context. In synthetic biology, it is used for attempts to create life from scratch (see very recent and excellent treatment of this matter by Fussenegger: http://onlinelibrary.wiley.com/doi/10.1002/anie.201609229/full ).
ItTherefore, it should be replaced with “Changing the genetic code rules in single proteins expressed in vivo” or similar.
The term “top-down approach” should be replaced by “Proteome wide insertions of NCAA” or “Genome wide codon reassignments for NCAA incorprations in all cellular proteins” or something similar.
In chapter 3, you talk about inhibitory effects of NCAA presence in cells and cellular proteins, isn’t it?
In section 3.4 the importance of metabolic engineering and its coupling with the GCE should be also elaborated (see: (a)R. A. Mehl, J. C. Anderson, S. W. Santoro, L. Wang, A. B. Martin, D. S. King, D. M. Horn, P. G. Schultz, J. Am. Chem. Soc. 2003, 125, 935–939; (b) W. Ou, T. Uno, H.-P. Chiu, J. Grunewald, S. E. Cellitti, T. Crossgrove, X. Hao, Q. Fan, L. L. Quinn, P. Patterson, et al., Proc. Natl. Acad. Sci. 2011, 108, 10437–10442; (c) M. Ehrlich, M. J. Gattner, B. Viverge, J. Bretzler, D. Eisen, M. Stadlmeier, M. Vrabel, T. Carell, Chem. - A Eur. J. 2015, 21, 7701–7704; (d) Y. Ma, H. Biava, R. Contestabile, N. Budisa, M. L. Di Salvo, Molecules 2014, 19, 1004–1022; (e) M. Exner, T. Kuenzl, T. To, Z. Ouyang, S. Schwagerus, M. Hoesl, C. Hackenberger, M. Lensen, S. Panke, N. Budisa, ChemBioChem 2016).
In chapter 4, Xenobiology should be mentioned (see e.g.: Agostini F, Völler J-S, Koksch B, Acevedo-Rocha CG, Kubyshkin V, Budisa N: Xenobiology meets enzymology.Angew. Chem. Int. Ed. Engl. 2017, doi:10.1002/anie.201610129).
Given that these substantial (and essential) changes are done in the revised manuscript, I would be ready to review it again and recommend its publication.
Author Response
We thank the reviewer for the constructive suggestions in various aspects. First of all, we have redefined the focus of this review as “genetic code engineering” instead of “thawing the frozen accident”, as can be seen in the title, abstract and introduction. Second, some concepts have been clarified according to the reviewer’s comments. For example, in line 15, the main idea is about “genetic code engineering” instead of “genetic code modification,” and the term “genetic code engineering” is used elsewhere to replace “genetic code modification” in the remaining text. The discussion is not limited to incorporation of NCAAs proteome-wide, but also included sites-specific ones. Third, some essential information about genetic code is also included (line 24-30). However, since this review is not focusing on genetic code theories, readers are properly directed to other well written reviews. Fourth, we very much appreciate the idea of xenobiology and have incorporated it into the text (line 47-48, 234-240). Fifth, terminologies such as “top-down” and “bottom-up” have been revised properly (chapter 2.1 and 2.2). Sixth, for chapter 3, it mainly touches upon challenges in genetic code engineering and the sub-headers have been modified accordingly. Finally, the term “metabolic engineering” has been adopted with proper citations (line 107-109).
Reviewer 3 Report
review of Ms: Thawing the frozen landscape: Efforts and challenges regarding genetic code modification
authors: Lin, Yu and Chan
review by Herve Seligmann
This short minireview presents an overview of biotechnological methods to insert unusual/unnatural amino acids in proteins, in other words genetic code modifications.
I consider the review publishable as is, but suggest the following issues to be developed.
In my view, integrating a discussion of these topics would increase the value of this ms. The extent of these discussions should be at the discretion of the authors, but note that my overview of potentially relevant literature is far from extensive.
Please note that I could not see on the pdf provided figure 1.
Publications on the subject of this ms usually neglect to describe that many unusual
amino acids are naturally integrated in proteins. Basically, one should explain that biotechnology takes advantage of natural phenomena that occur spontaneously in cells.
These concern selenocystein (not mentioned in ms) and pyrolysine (mentioned in ms) for stop codon suppression in bacteria, and that many different (regular) amino acids are apparently frequently inserted at stop codons 'in nature', with a bias for amino acids coded by near cognate anticodons (lysine and glutamine, see Aerni H.R., Shifman M.A., Rogulina S., O'Donoghue P., Rinehart J. Revealing the amino acid composition of proteins within an expanded genetic code. Nucleic Acids Res. 2015;43:e8).
This was also confirmed also for human mitochondria
(Seligmann H 2016 Unbiased Mitoproteome Analyses Confirm Non-canonical RNA, Expanded Codon Translations. Comput Struct Biotechnol J 14, 391-403; Seligmann H 2016 Natural chymotrypsin-like-cleaved human mitochondrial peptides confirm tetra-, pentacodon, non-canonical RNA translations. Biosystems 147, 78-93; Seligmann H 2016 Natural mitochondrial proteolysis confirms transcription systematically exchanging/deleting nucleotides, peptides coded by expanded codons. J Theor Biol 414, 76-90; Seligmann H 2016 Translation of mitochondrial swinger RNAs according to tri-, tetra- and pentacodons. Biosystems 140, 38-48;
Seligmann H 2015 Codon expansion and systematic transcriptional deletions produce tetra-, pentacoded mitochondrial peptides. J Theor Biol 387, 154-165).
In mitochondria, predicted occurrences of stop-suppressor tRNAs (H. Seligmann Pathogenic mutations in antisense mitochondrial tRNAs J. Theor. Biol., 269 (2011), pp. 287–296; H. Seligmann Avoidance of antisense, antiterminator tRNA anticodons in vertebrate mitochondria Biosystems, 101 (2010), pp. 42–50; H. Seligmann Pocketknife tRNA hypothesis: anticodons in mammal mitochondrial tRNA side-arm loops translate proteins? Biosystems, 113 (2013), pp. 165–176; H. Seligmann Putative anticodons in mitochondrial tRNA sidearm loops: pocketknife tRNAs? J. Theor. Biol., 340 (2014), pp. 155–163)
coevolve with predicted translation of stops (E. Faure, L. Delaye, S. Tribolo, A. Levasseur, H. Seligmann, R.M. Barthélémy Probable presence of an ubiquitous cryptic mitochondrial gene on the antisense strand of the cytochrome oxidase I gene Biol. Direct, 6 (2011), p. 56; H. Seligmann Two genetic codes, one genome: frameshifted primate mitochondrial genes code for additional proteins in presence of antisense antitermination tRNAs Biosystems, 105 (2011), pp. 271–285; H. Seligmann An overlapping genetic code for frameshifted overlapping genes in Drosophila mitochondria: antisense antitermination tRNAs UAR insert serine J. Theor. Biol., 298 (2012), pp. 51–76; Barthélémy RM, Seligmann H. Cryptic tRNAs in chaetognath mitochondrial genomes. Comput Biol Chem. 2016 Jun;62:119-32.)
Some mitochondria require systematic stop codon reassignment to produce part of the regular proteins necessary for mitochondrial metabolism (R.D. Russell, A.T. Beckenbach Recoding of translation in turtle mitochondrial genomes: programmed frameshift mutations and evidence of a modified genetic code J. Mol. Evol., 67 (2008), pp. 682–695; H. Seligmann Overlapping genetic codes for overlapping frameshifted genes in Testudines, and Lepidochelys olivacea as special case Comput. Biol. Chem., 41 (2012), pp. 18–34).
Reconstructing a realistic phylogeny of mitochondrial genetic codes requires assuming dual stop/amino acid coding for stops (H. Seligmann Phylogeny of genetic codes and punctuation codes within genetic codes Biosystems, 129 (2015), pp. 36–43)
Stop codon reassignment also occurs in eukaryotes (i.e. M. Adachi, A. Cavalcanti Tandem stop codons in ciliates that reassign stop codons J. Mol. Evol., 68 (2009), pp. 424–431; O. Beznoskova, S. Gunisova, L.S. Valasek Rules of UGA-N decoding by near-cognate tRNAs and analysis of readthrough on short uORFs in yeast RNA, 22 (2016), pp. 456–466)
Translation according to expanded codons is also a natural phenomenon, discovered in the early 1970s, and that is more common than believed. It includes translation by tRNAs with expanded anticodons (L.F. Landweber Custom codons come in threes, fours, and fives Chem. Biol., 9 (2002), p. 143; M. O'Connor tRNA hopping: effects of mutant tRNAs Biochim. Biophys. Acta, 1630 (2003), pp. 41–46; D.L. Riddle, J. Carbon Frameshift suppression: a nucleotide addition in the anticodon of a glycine transfer RNA Nat. New Biol., 242 (1973), pp. 230–234; J.C. Anderson, T.J. Magliery, P.G. Schultz Exploring the limits of codon and anticodon size Chem. Biol., 9 (2002), pp. 237–244; T.J. Magliery, J.C. Anderson, P.G. Schultz Expanding the genetic code: selection of efficient suppressors of four-base codons and identification of “shifty” four-base codons with a library approach in Escherichia coli J. Mol. Biol., 307 (2001), pp. 755–769; B. Moore, B.C. Persson, C.C. Nelson, R.F. Gesteland, J.F. Atkins Quadruplet codons: implications for code expansion and the specification of translation step size J. Mol. Biol., 298 (2000), pp. 195–209; G.E. Sroga, F. Nemoto, Y. Kuchino, G.R. Bjork Insertion (sufB) in the anticodon loop or base substitution (sufC) in the anticodon stem of tRNA(Pro)2 from Salmonella typhimurium induces suppression of frameshift mutations. Nucleic Acids Res., 20 (1992), pp. 3463–3469; T.M. Tuohy, S. Thompson, R.F. Gesteland, J.F. Atkins Seven, eight and nine-membered anticodon loop mutants of tRNA(2Arg) which cause +1 frameshifting. Tolerance of DHU arm and other secondary mutations. J. Mol. Biol., 228 (1992), pp. 1042–1054).
In mitochondria, predicted tRNAs with expanded anticodons (H. Seligmann Undetected antisense tRNAs in mitochondrial genomes? Biol. Direct, 5 (2010), p. 39) coevolve with predicted protein coding regions coded by expanded codons (H. Seligmann Putative mitochondrial polypeptides coded by expanded quadruplet codons, decoded by antisense tRNAs with unusual anticodons Biosystems, 110 (2012), pp. 84–106; H. Seligmann Pocketknife tRNA hypothesis: anticodons in mammal mitochondrial tRNA side-arm loops translate proteins? Biosystems, 113 (2013), pp. 165–176; H. Seligmann Putative anticodons in mitochondrial tRNA sidearm loops: pocketknife tRNAs? J. Theor. Biol., 340 (2014), pp. 155–163)
and with the organisms life history, suggesting that expanded anticodons are an adaptation to translation at high temperatures, where regular codon/anticodon interactions are thermodynamically unstable (H. Seligmann, A. Labra Tetracoding increases with body temperature in Lepidosauria. Biosystems, 114 (2013), pp. 155–163)
This approach also suggests that the genetic code's ancestor, presumably functioning at high temperatures (M. Di Giulio The universal ancestor lived in a thermophilic or hyperthermophilic environment J. Theor. Biol., 203 (2000), pp. 203–213), had expanded codons (P.V. Baranov, M. Venin, G. Provan Codon size reduction as the origin of the triplet genetic code PLoS One, 4 (2009), p. e5708).
The hypothesis of an ancestral mitochondrial genetic code with expanded codons also fits error minimization principles (D.L. Gonzalez, S. Giannerini, R. Rosa On the origin of the mitochondrial genetic code: towards a unified mathematical framework for the management of genetic information Nat. Preced. (2012) http://precedings.nature.com/documents/7136/version/1).
Numerous peptides with unknown function matching translation according to expanded codons occur within human mitochondrial proteomic data (H. Seligmann Codon expansion and systematic transcriptional deletions produce tetra-, pentacoded mitochondrial peptides J. Theor. Biol., 387 (2015), pp. 154–165; H. Seligmann Translation of mitochondrial swinger RNAs according to tri-,tetra- and pentacodons Biosystems, 140 (2016), pp. 38–48; H. Seligmann Natural chymotrypsin-like-cleaved human mitochondrial peptides confirm tetra-, pentacodon, non-canonical RNA translations Biosystems, 147 (2016), pp. 78–93; H. Seligmann Unbiased mitoproteome analyses confirm non-canonical RNA, expanded codon translations Comput. Struct. Biotech. J., 14 (2016), pp. 391–403)
Author Response
We thank the reviewer for providing suggestions in a wide range of topics to increase the value and extend the discussions of our manuscript. In particular, we have talked about alternative genetic code in nature in the introduction (line 24-35). We have properly selected some aspects from these suggestions and integrated them in the revised version. For example, the reviewer points out the use of tetra- and pentacodon in mitochondria. If the mechanism of this phenomenon is revealed, it may have the potential to serve as a tool for genetic code engineering (line 80-83). We did not include all the suggestions as provided by the reviewer because our ultimate focus is on genetic code engineering.
Round 2
Reviewer 2 Report
Manuscript is substantially improved. Please polish english lenaguage.
Author Response
We thank the reviewer for reading our revised manuscript carefully. We have polished the language to our best in the current version.